# Park: An Open Platform for Learning-Augmented Computer Systems

**Hongzi Mao** [1]   **Parimarjan Negi** [1]   **Akshay Narayan** [1]   **Hanrui Wang** [1]   **Jiacheng Yang** [1]   **Haonan Wang** [1]
**Ryan Marcus** [1]   **Ravichandra Addanki** [1]   **Mehrdad Khani** [1]   **Songtao He** [1]   **Vikram Nathan** [1]   **Frank Cangialosi** [1]
**Shaileshh Bojja Venkatakrishnan** [1]   **Wei-Hung Weng** [1]   **Song Han** [1]   **Tim Kraska** [1]   **Mohammad Alizadeh** [1]

## Abstract

We present Park, a platform for researchers to experiment with Reinforcement Learning (RL) for computer systems. Using RL for improving the performance of systems has a lot of potential, but is also in many ways very different from, for example, using RL for games. Thus, in this work we first discuss the unique challenges RL for systems has, and then propose Park an open extensible platform, which makes it easier for ML researchers to work on systems problems. Currently, Park consists of 12 real world system-centric optimization problems with one common easy to use interface. Finally, we present the performance of existing RL approaches over those 12 problems and outline potential areas of future work.

## 1. Introduction

Deep reinforcement learning (RL) has emerged as a general and powerful approach to sequential decision making problems in recent years. However, real-world applications of deep RL have thus far been limited. The successes, while impressive, have largely been confined to controlled environments, such as complex games (Mnih et al., 2015; Silver et al., 2018; Tian et al., 2017; OpenAI, 2018; Vinyals et al., 2019) or simulated robotics tasks (Rajeswaran et al., 2017; OpenAI et al., 2018; Hwangbo et al., 2019). This paper concerns applications of RL in computer systems, a relatively unexplored domain where RL could provide significant real-world benefits.

Computer systems are full of sequential decision-making tasks that can naturally be expressed as Markov decision processes (MDP). Examples include caching (operating systems), congestion control (networking), query optimization (databases), scheduling (distributed systems), and more (§2).

[1]MIT Computer Science and Artificial Intelligence Laboratory. Correspondence to: Hongzi Mao <hongzi@csail.mit.edu>.

*Reinforcement Learning for Real Life (RL4RealLife) Workshop in the 36th International Conference on Machine Learning*, Long Beach, California, USA, 2019. Copyright 2019 by the author(s).

Since real-world systems are difficult to model accurately, state-of-the-art systems often rely on human-engineered heuristic algorithms that can leave significant room for improvement (Mirhoseini et al., 2017). Further, these algorithms can be complex (e.g., a commercial database query optimizer involves hundreds of rules (Begoli et al., 2018)), and are often difficult to adapt across different systems and operating environments (Mao et al., 2018; Marcus et al., 2019) (e.g., different workloads, different distribution of data in a database, etc.). Furthermore, unlike control applications in physical systems, most computer systems run in software on readily-available commodity machines. Hence the cost of experimentation is much lower than physical environments such as robotics, making it relatively easy to generate abundant data to explore and train RL models. This mitigates (but does not eliminate) one of the drawbacks of RL approaches in practice — their high sample complexity (Amodei & Hernandez, 2018; Sun et al., 2018). The easy access to training data and the large potential benefits have attracted a surge of recent interest in the systems community in applying RL to various problems (Mao et al., 2016; 2018; Jay et al., 2018; Krishnan et al., 2018; Marcus et al., 2019; Boyan & Littman, 1994; Mao et al., 2017; Mirhoseini et al., 2017; 2018; Gao et al., 2018; Li, 2019).

From a machine learning perspective, computer systems present many challenging problems for RL. The landscape of decision-making problems in systems is vast, ranging from centralized control problems (e.g., a scheduling agent responsible for an entire computer cluster) to distributed multi-agent problems where multiple entities with partial information collaborate to optimize system performance (e.g., network congestion control with multiple connections sharing bottleneck links). Further, the control tasks manifest at a variety of timescales, from fast, reactive control systems with sub-second response-time requirements (e.g., admission/eviction algorithms for caching objects in memory) to longer term planning problems that consider a wide range of signals to make decisions (e.g., VM allocation/placement in cloud computing). Importantly, computer systems give rise to new challenges for learning algorithms that are not common in other domains (§3). Examples of these challenges include time-varying state or action spaces (e.g.,

dynamically varying number of jobs and machines in a computer cluster), structured data sources (e.g., graphs to represent data flow of jobs or a network's topology), and highly stochastic environments (e.g., random time-varying workloads). These challenges present new opportunities for designing RL algorithms. For example, motivated by applications in networking and queuing systems, recent work (Mao et al., 2019) developed new general-purpose control variates for reducing variance of policy gradient algorithms in "input-driven" environments, in which the system dynamics are affected by an exogenous, stochastic process.

Despite these opportunities, there is relatively little work in the machine learning community on algorithms and applications of RL in computer systems. We believe a primary reason is the lack of good benchmarks for evaluating solutions, and the absence of an easy-to-use platform for experimenting with RL algorithms in systems. Conducing research on learning-based systems currently requires significant expertise to implement solutions in real systems, collect suitable real-world traces, and evaluate solutions rigorously. The primary goal of this paper is to lower the barrier of entry for machine learning researchers to innovate in computer systems.

We present Park, an open, extensible platform that presents a common RL interface to connect to a suite of 12 computer system environments (§3.3). These representative environments span a wide variety of problems across networking, databases, and distributed systems, and range from centralized planning problems to distributed fast reactive control tasks. In the backend, the environments are powered by both real systems (in 7 environments) and high fidelity simulators (in 5 environments). For each environment, Park defines the MDP formulation, e.g., events that triggers an MDP step, the state and action spaces and the reward function. This allows researchers to focus on the core algorithmic and learning challenges, without having to deal with low-level system implementation issues. At the same time, Park makes it easy to compare different proposed learning agents on a common benchmark, similar to how OpenAI Gym (Brockman et al., 2016) has standardized RL benchmarks for robotics control tasks. Finally, Park defines a RPC interface (Srinivasan, 1995) between the RL agent and the backend system, making it easy to extend to more environments in the future.

We benchmark the 12 systems in Park with both RL methods and existing heuristic baselines (§5). The experiments benchmark the training efficiency and the eventual performance of RL approaches on each task. The empirical results are mixed: RL is able to outperform state-of-the-art baselines in several environments where researchers have developed problem-specific learning methods; for many other systems, RL has yet to consistently achieve robust performance. We open-source Park as well as the RL agents and baselines in https://github.com/park-project/park.

## 2. Sequential Decision Making Problems in Computer Systems

Sequential decision making problems manifest in a variety of ways across computer systems disciplines. These problems span a multi-dimensional space from centralized vs. multi-agent control to reactive, fast control loops vs. long-term planning. In this section, we overview a sample of problems from each discipline and how to formulate them as MDPs. Appendix A provides further examples and a more formal description of the MDPs that we have implemented in Park.

**Networking.** Computer network problems are fundamentally distributed, since they interconnect independent users. One example is congestion control, where hosts in the network must each determine the rate to send traffic, accounting for both the capacity of the underlying network infrastructure and the demands of other users of the network. Each network connection has an agent (typically at the sender side) setting the sending rate based on how previous packets were acknowledged. This component is crucial for maintaining a large throughput and low delay.

Another example at the application layer is bitrate adaptation in video streaming. When streaming videos from content provider, each video is divided into multiple chunks. At watch time, an agent decides the bitrate (affecting resolution) of each chunk of the video based on the network (e.g., bandwidth and latency measurements) and video characteristics (e.g., type of video, encoding scheme, etc.). The goal is to learn a policy that maximizes the resolution while minimizing chance of stalls (when slow network cannot download a chunk fast enough).

**Databases.** Databases seek to efficiently organize and retrieve data in response to user requests. To efficiently organize data, it is important to index, or arrange, the data to suit the retrieval patterns. An indexing agent could observe query patterns and accordingly decide how to best structure, store, and over time, re-organize the data.

Another example is query optimization. Modern query optimizers are complex heuristics which use a combination of rules, handcrafted cost models, data statistics, and dynamic programming, with the goal to re-order the query operators (e.g., joins, predicates) to ultimately lower the execution time. Unfortunately, existing query optimizers do not improve over time and do not learn from mistakes. Thus, they are an obvious candidate to be optimized through RL (Marcus et al., 2019). Here, the goal is to learn a query optimization policy based on the feedback from optimizing and running a query plan.

**Distributed systems.** Distributed systems handle computations that are too large to fit on one computer; for example, the Spark framework for big-data processing computes results across data stored on multiple computers (Zaharia

et al., 2012). To efficiently perform such computations, a job scheduler decides how the system should assign compute and memory resources to jobs to achieve fast completion times. Data processing jobs often have complex structure (e.g., Spark jobs are structured as dataflow graphs, Tensorflow models are computation graphs). The agent in this case observes a set of jobs and the status of the compute resources (e.g., how each job is currently assigned). The action decides how to place jobs onto compute resources. The goal is to complete the jobs as soon as possible.

**Operating systems.** Operating systems seek to efficiently multiplex hardware resources (compute, memory, storage) amongst various application processes. One example is providing a memory hierarchy: computer systems have a limited amount of fast memory and relatively large amounts of slow storage. Operating systems provide caching mechanisms which multiplex limited memory amongst applications which achieve performance benefits from residency in faster portions of the cache hierarchy. In this setting, an RL agent can observe the information of both the existing objects in the cache and the incoming object; it then decides whether to admit the incoming object and which stale objects to evict from the cache. The goal is to maximize the cache hit rate (so that more application reads occur from fast memory) based on the access pattern of the objects.

Another example is CPU power state management. Operating systems control whether the CPU should run at an increased clock speed and boost application performance, or save energy with at a lower clock speed. An RL agent can dynamically control the clock speed based on the observation of how each application is running (e.g., is an application CPU bound or network bound, is the application performing IO tasks). The goal is to maintain high application performance while reducing the power consumption.

## 3. RL for Systems Characteristics and Challenges

In this section, we explain the unique characteristics and challenges that often prevent off-the-shelf RL methods from achieving strong performance in different computer system problems. Admittedly, each system has its own complexity and contains special challenges. Here, we primarily focus on the common challenges that arise across many systems in different stages of the RL design pipeline.

### 3.1. State-action Space

**The needle-in-the-haystack problem.** In some computer systems, the majority of the state-action space presents little difference in reward feedback for exploration. This provides no meaningful gradient during RL training, especially in the beginning, when policies are randomly initialized. Network

congestion control is a classic example: even in the simple case of a fixed-rate link, setting the sending rate above the available network bandwidth saturates the link and the network queue. Then, changes in the sending rate above this threshold result in an equivalently bad throughput and delay, leading to constant, low rewards. To exit this bad state, the agent must set a low sending rate for multiple *consecutive* steps to drain the queue before receiving any positive reward. Random exploration is not effective at learning this behavior because any random action can easily overshadow several good actions, making it difficult to distinguish good action sequences from bad ones. Circuit design is another example: when *any* of the circuit components falls outside the operating region (the exact boundary is unknown before invoking the circuit simulator), the circuit cannot function properly and the environment returns a constant bad reward. As a result, exploring these areas provides little gradient for policy training.

In these environments, using domain-knowledge to confine the search space helps to train a strong policy. For example, we observed significant performance improvements for network congestion control problems when restricting the policy (see also Figure 4(d)). Also, environment-specific reward shaping (Ng et al., 1999) or bootstrapping from existing policies (Silver et al., 2016; Hester et al., 2017) can improve policy search efficiency.

**Representation of state-action space.** When designing RL methods for problems with complex structure, properly encoding the state-action space is the key challenge. In some systems, the action space grows exponentially large as the problem size increases. For example, in switch scheduling, the action is a bijection mapping (a matching) between input and output ports — a standard 32-port would have 32! possible matching. Encoding such a large action space is challenging and makes it hard to use off-the-shelf RL agents. In other cases, the size of the action space is constantly changing over time. For example, a typical problem is to map jobs to machines. In this case, the number of possible mappings and thus, actions increases with the number of new jobs in the system.

Unsurprisingly, domain specific representations that capture inherent structure in the state space can significantly improve training efficiency and generalization. For example, Spark jobs, Tensorflow components, and circuit design are to some degree dataflow graphs. For these environments, leveraging Graph Convolutional Neural Networks (GCNs) (Kipf & Welling, 2016) rather than LSTMs can significantly improves generalization (see Table 1). However, finding the right representation for each problem is a central challenge, and for some domains, e.g., query optimization, remains largely unsolved.

| | GCN direct | GCN transfer | LSTM direct | LSTM transfer | Random |
|---|---|---|---|---|---|
| CIFAR-10 (Krizhevsky & Hinton, 2010) | **1.73**±0.41 | **1.81**±0.39 | **1.78**±0.38 | 1.97±0.37 | 2.15±0.39 |
| Penn Tree Bank (Marcus et al., 1993) | **4.84**±0.64 | **4.96**±0.63 | 5.09±0.63 | 5.28±0.6 | 5.42±0.57 |
| NMT (Bahdanau et al., 2014) | **1.98**±0.55 | **2.07**±0.51 | 2.16±0.56 | 2.88±0.66 | 2.47±0.48 |

*Table 1.* Generalizability of GCN and LSTM state representation in the Tensorflow device placement environment. The numbers are average runtime in seconds. ± spans one standard deviation. Bold font indicate the runtime is within 5% of the best runtime. "Transfer" means testing on unseen models in the dataset.

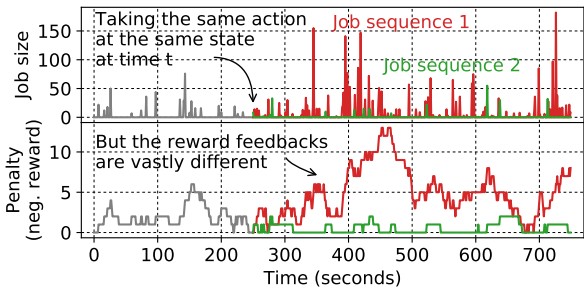

*Figure 1.* Illustrative example of load balancing showing how different instances of a stochastic input process can have vastly different rewards. After time $t$, we sample two job arrival sequences from a Poisson process. Figure adopted from Mao et al. (2018).

### 3.2. Decision Process

**Stochasticity in MDP causing huge variance.** Queuing systems environments (e.g., job scheduling, load balancing, cache admission) have dynamics partially dictated by an exogenous, stochastic *input process*. Specifically, their dynamics are governed not only by the decisions made within the system, but also the arrival process that brings work (e.g., jobs, packets) into the system. In these environments, the stochasticity in the input process causes huge variance in the reward.

For illustration, consider the load balancing example in Figure 1. If the arrival sequence after time $t$ consists of a burst of large jobs (e.g., job sequence 1), the job queue will grow and the agent will receive low rewards. In contrast, a stream of lightweight jobs (e.g., job sequence 2) will lead to short queues and large rewards. The problem is that this difference in reward is independent of the action at time $t$; rather, it is caused purely by the randomness in the job arrival process. In these environments, the agents cannot tell whether two reward feedbacks differ due to disparate input processes, or due to the quality of the actions. As a result, standard methods for estimating the value of an action suffer from high variance.

Prior work proposed an input-dependent baseline that effectively reduces the variance from the input process Mao et al. (2019). Figure 5 in Mao et al. (2019) shows the policy improvement when using input-dependent baselines in the load-balancing and adaptive video streaming environments. However, the proposed training implementations ("multi-value network" and "meta baseline") are tailored for policy gradient methods and require the environments to have a repeatable

input process (e.g., in simulation, or real systems with controllable input sequence). Thus, coping with input-driven variance remains an open problem for value-based RL methods and for environments with uncontrollable input processes.

**Infinite horizon problems.** In practice, production computer systems (e.g., Spark schedulers, load balancers, cache controllers, etc.) are long running and host services indefinitely. This creates an infinite horizon MDP (Baxter & Bartlett, 2001) that prevents the RL agents from performing episodic training. In particular, this creates difficulties for bootstrapping a value estimation since there is no terminal state to easily assign a known target value. Moreover, the discounted total reward formulation in the episodic case might not be suitable — an action in a long running system can have impact beyond a fixed discounting window. For example, scheduling a large job on a slow server blocks future small jobs (affecting job runtime in the rewards), no matter whether the small jobs arrive immediately after the large job or much farther in the future over the course of the lifetime of the large job. Average reward RL formulations can be a viable alternative in this setting (see §10.3 in Sutton & Barto (2017) for an example).

### 3.3. Simulation-Reality Gap

Unlike training RL in simulation, robustly deploying a trained RL agent or directly training RL on an actual running computer systems has several difficulties. First, discrepancies between simulation and reality prevent direct generalization. For example, in database query optimization, existing simulators or query planners use offline cost models to predict query execution time (as a proxy for the reward). However, the accuracy of the cost model quickly degrades as the query gets more complex due to both variance in the underlying data distribution and system-specific artifacts (Leis et al., 2015).

Second, interactions with some real systems can be slow. In adaptive video streaming, for example, the agent controls the bitrate for each chunk of a video. Thus, the system returns a reward to the agent only after a video chunk is downloaded, which typically takes a few seconds. Naively using the same training method from simulation (as in Figure 4(a)) would take a single-threaded agent more than 10 years to complete training in reality.

Finally, live training or directly deploying an agent from simulation can degrade the system performance. Figure 2

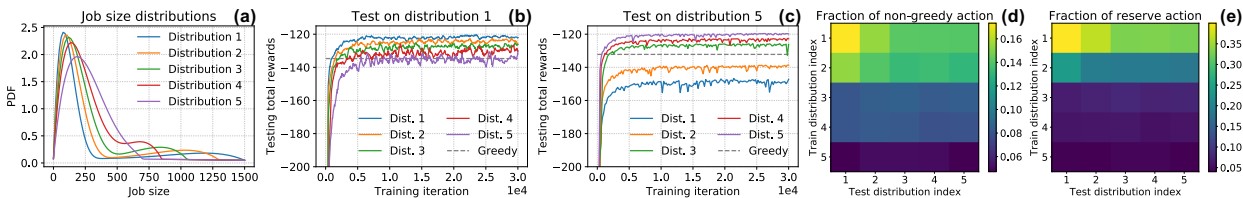

*Figure 2.* Demonstration of the gap between simulation and reality in the load balancing environment. (a) Distribution of job sizes in the training workload. (b, c) Testing agents on a particular distribution. An agent trained with distribution 5 is more robust than one trained with distribution 1. (d, e) A "reservation" policy that keeps a server empty for small jobs. Such a policy overfits distribution 1 and is not robust to workload changes.

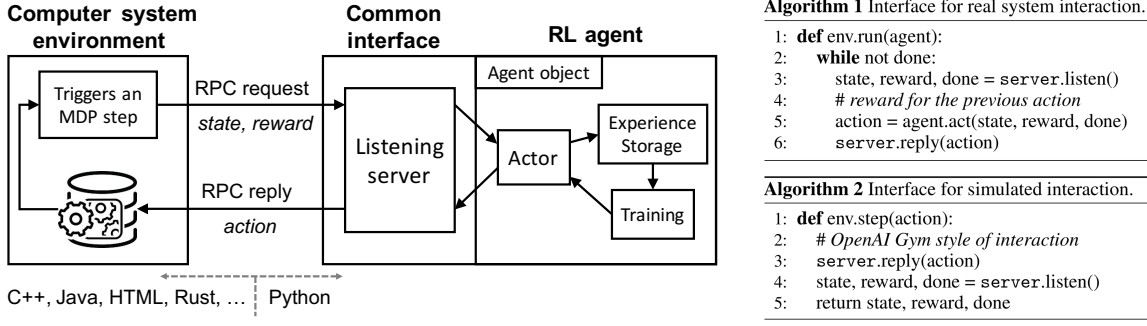

*Figure 3.* Park architects an RL-as-a-service design paradigm. The computer system connects to an RL agent through a canonical request/response interface, which hides the system complexity from the RL agent. Algorithm 1 describes a cycle of the system interaction with the RL agent. By wrapping with an agent-centric environment in Algorithm 2, Park's interface also supports OpenAI Gym (Brockman et al., 2016) like interaction for simulated environments.

describes a concrete example for load balancing. When training with a bimodal distribution job sizes, the RL agent learns to reserve a certain server for small jobs to process them quickly. However, when the distribution changes, blindly reserving a server wastes compute resource and reduces system throughput. Therefore, to deploy training algorithms online, these problems require RL to train robust policies that ensure safety (García & Fernández, 2015; Achiam et al., 2017; Kang et al., 2018).

### 3.4. Understandability over Existing Heuristics

As in other areas of ML, interpretability plays an important role in making learning techniques practical. However, in contrast to perception-based problems or games, for system problems, many reasonable good heuristics exist. For example, every introductory course to computer science features a basic scheduling algorithm such as FIFO. These heuristics are often easy to understand and to debug, whereas a learned approach is often not. Hence, making learning algorithms in systems as debuggable and interpretable as existing heuristics is a key challenge. Here, a unique opportunity is to build hybrid solutions, which combine learning-based techniques with traditional heuristics. Existing heuristics can not only help to bootstrap certain problems, but also help with safety and generalizability. For example, a learned scheduling algorithm could fall back to a simple heuristic

if it detects that the input distribution significantly drifted.

## 4. The Park Platform

Park follows a standard request-response design pattern. The backend system runs continuously and periodically send requests to the learning agent to take control actions. To connect the systems to the RL agents, Park defines a common interface and hosts a server that listens for requests from the backend system. The backend system and the agent run on different processes (which can also run on different machines) and they communicate using remote procedure calls (RPCs). This design essentially structures RL as a service. Figure 3 provides an overview of Park.

**Real system interaction loop.** Each system defines its own events to trigger an MDP step. At each step, the system sends an RPC request that contains a current observation of the state and a reward corresponding to the *last* action. Upon receiving the request, the Park server invokes the RL agent. For simplicity, we pass an instance of the agent object directly to the server at initialization. However, the agent can also live in a different process (e.g., if we need to live-update the agent's implementation while the system is still running). Park provides the agent with all the information provided by the environment. The implementation of the agent is up to the users (e.g., feature extraction, training process, inference

| Environment | Type | State space | Action space | Reward | Step time | Challenges (§3) |
|---|---|---|---|---|---|---|
| Adaptive video streaming | Real/sim | Past network throughput measurements, playback buffer size, portion of unwatched video | Bitrate of the next video chunk | Combination of resolution and stall time | Real: ~3s Sim: ~1ms | Input-driven variance, slow interaction time |
| Spark cluster job scheduling | Real/sim | Cluster and job information as features attached to each node of the job DAGs | Node to schedule next | Runtime penalty of each job | Real: ~5s Sim: ~5ms | Input-driven variance, state representation, infinite horizon, reality gap |
| SQL database query optimization | Real | Query graph with predicate and table features on nodes, join attributes on edges | Edge to join next | Cost model or actual query time | ~5s | State representation, reality gap |
| Network congestion control | Real | Throughput, delay and packet loss | Congestion window and pacing rate | Combination of throughput and delay | ~10ms | Sparse space for exploration, safe exploration, infinite horizon |
| Network active queue management | Real | Past queuing delay, enqueue/dequeue rate | Drop rate | Combination of throughput and delay | ~50ms | Infinite horizon, reality gap |
| Tensorflow device placement | Real/sim | Current device placement and runtime costs as features attached to each node of the job DAGs | Updated placement of the current node | Penalty of runtime and invalid placement | Real: ~2s Sim: ~10ms | State representation, reality gap |
| Circuit design | Sim | Circuit graph with component ID, type and static parameters as features on the node | Transistor sizes, capacitance and resistance of each node | Combination of bandwidth, power and gain | ~2s | State representation, sparse space for exploration |
| CDN memory caching | Sim | Object size, time since last hit, cache occupancy | Admit/drop | Byte hits | ~2ms | Input-driven variance, infinite horizon, safe exploration |
| Multi-dim database indexing | Real | Query workload, stored data points | Layout for data organization | Query throughput | ~30s | State/action representation, infinite horizon |
| Account region assignment | Sim | Account language, region of request, set of linked websites | Account region assignment | Serving cost in the future | ~1ms | State/action representation |
| Server load balancing | Sim | Current load of the servers and the size of incoming job | Server ID to assign the job | Runtime penalty of each job | ~1ms | Input-driven variance, infinite horizon, safe exploration |
| Switch scheduling | Sim | Queue occupancy for input-output port pairs | Bijection mapping from input ports to output ports | Penalty of remaining packets in the queue | ~1ms | Action representation |

*Table 2.* Overview of the computer system environments supported by Park platform.

methods). Once the agent returns an action, the server replies back to the system. In Figure 3, Algorithm 1 depicts this interaction process.

Notice that invoking the agent incurs a physical delay for the RPC response from the server. Depending on the underlying implementation, the system may or may not wait synchronously during this delay. For a real system using non-blocking RPC, the state observation received by the agent can be stale (which would typically not occur in simulation). On the other hand, if the system makes blocking RPC requests, then taking a long time to compute an action (e.g., while performing MCTS search (Silver et al., 2018)) can degrade the system performance. Designing high-performance RL training or inference agents in a real computer system should explicitly take this delay factor into account.

**Wrapper for simulated interaction.** By wrapping the request-response interface with a shim layer, Park also supports an "agent-centric" style of interaction advocated by OpenAI Gym (Brockman et al., 2016). In Figure 3, Algorithm 2 outlines this option in simulated system environments. The agent explicitly steps the environment forward by

sending the action to the underlying system through the RPC response. The interface then waits on the RPC server for the next action request. With this interface, we can directly reuse existing off-the-shelf RL training implementations benchmarked on Gym (Dhariwal et al., 2017).

**Scalability.** The common interface allows multiple instances of a system environment to run concurrently. These systems can generate the experience in parallel to speed up RL training. As a concrete example, to implement IMPALA (Espeholt et al., 2018) style of distributed RL training, the interface takes multiple actor instance at initialization. Each actor corresponds to an environment instance. When receiving an RPC request, the interface then uses the RPC request ID to route the request to the corresponding actor. The actor reports the experience to the learner (globally maintained for all agents) when the experience buffer reaches the batch size for training and parameter updating.

**Environments.** Table 2 provides an overview of 12 environments that we have implemented in Park. Appendix A contains the detailed descriptions of each problem, its MDP definition, and explanations of why RL could provide

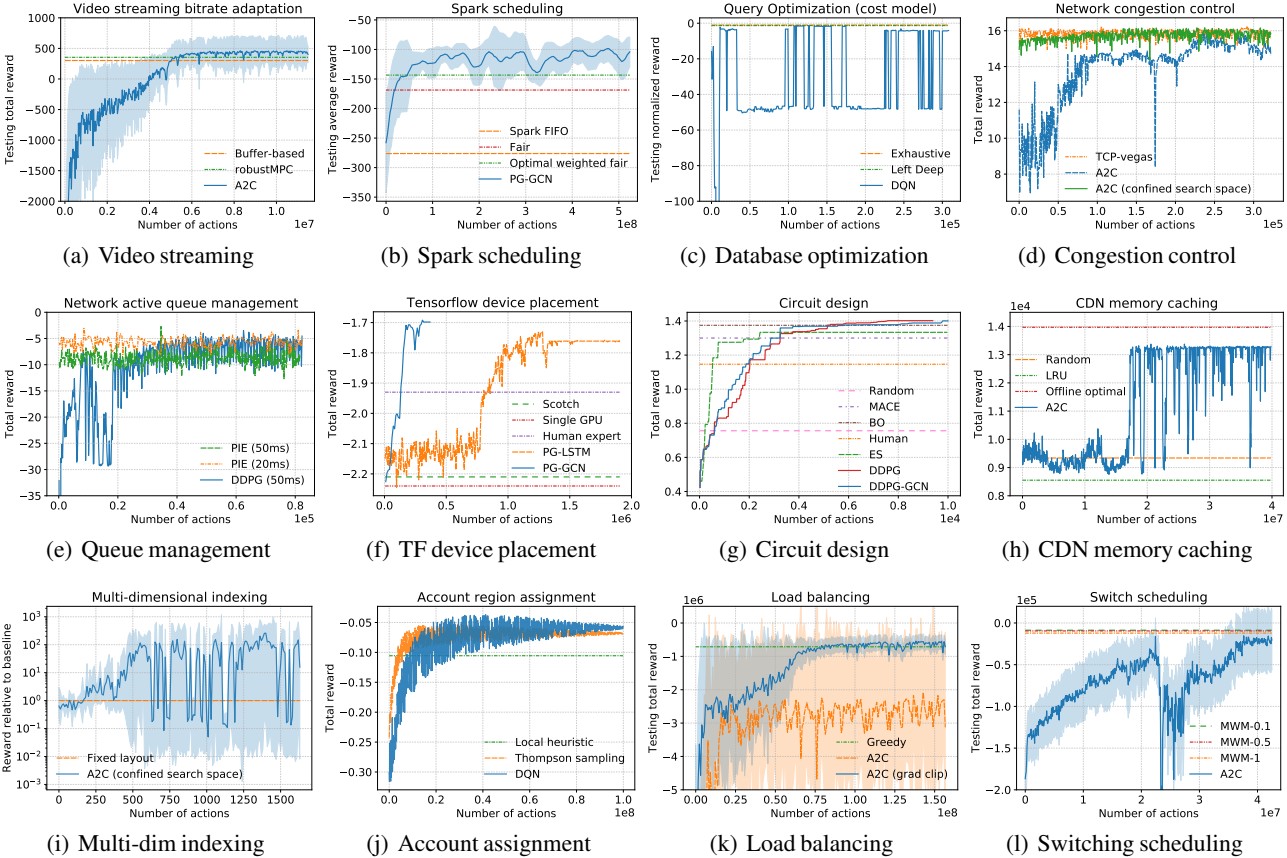

*Figure 4.* Benchmarks of the existing standard RL algorithms on Park environments. In y-axes, "testing" means the agents are tested with unseen settings in the environment (e.g., newly sampled workload unseen during training, unseen job patterns to schedule, etc.). The heuristic or optimal policies are provided as comparison.

benefits in each environment. Seven of the environments use real systems in the backend (see Table 2). For the remaining five environments, which have well-understood dynamics, we provide a simulator to facilitate easier setup and faster RL training. For these simulated environments, Park uses real-world traces to ensure that they mimic their respective real-world environments as faithfully as possible. For example, for the CDN memory caching environment, we use an open dataset containing 500 million requests, collected from a public CDN serving top-ten US websites (Berger, 2018). Given the request pattern, precisely simulating the dynamics of the cache (hits and evictions) is straightforward.

**Extensibility.** Adding a new system environment in Park is straightforward. For a new system, it only needs to specify (1) the state-action space definition (e.g., tensor, graph, powerset, etc.), (2) the event to trigger an MDP step, at which it sends an RPC request and (3) the function to calculate the reward feedback. From the agent's perspective, as long as the state-action space remains similar, it can use the same RL algorithm for the new environment. The common interface decouples the development of an RL agent from the complexity of the underlying system implementations.

## 5. Benchmark Experiments

We train the agents on the system environments in Park with several existing RL algorithms, including DQN (Mnih et al., 2015), A2C (Mnih et al., 2016), Policy Gradient (Sutton et al., 1999) and DDPG (Lillicrap et al., 2015). When available, we also provide the existing heuristics and the optimal policy for comparison. The details of hyperparameter settings, agent architecture and system configurations are in Appendix B. Figure 4 shows the experiment results. As a sanity check, the performance of the RL policy improves over time in all environments.

**Room for improvement.** We highlight system environments that potentially have large room for performance improvement. For database optimization in Figure 4(c), RL methods that make one-shot decisions, such as DQN, do not converge to a stable policy; combining with explicit search (Marcus et al., 2019) may improve the RL performance. In network congestion control, random exploration is inefficient to search the large state space that provides little reward gradient. Confining the search space with domain knowledge significantly improves learning efficiency in Fig-

ure 4(d) (more details in Appendix B.2). For Tensorflow device placement in Figure 4(f), using graph convolutional neural networks (GCNs) (Kipf & Welling, 2016) for state encoding is natural to the problem setting and allows the RL agent to learn more than 5 times faster than using LSTM encodings (Mirhoseini et al., 2018). Using more efficient encoding may improve the performance and generalizability further.

For some of the environments, we were forced to simplify the task to make it feasible to apply standard RL algorithms. Specifically, in CDN memory caching (Figure 4(h)), we only use a small 1MB cache (typical CDN caches are over a few GB); a large cache causes the reward (i.e., cache hit/miss) for an action to be significantly delayed (until the object is evicted from the cache, which can take hundreds of thousands of steps in large caches) (Berger, 2018). For account region assignment in Figure 4(j), we only allocate an account at initialization (without further migration). Active migration at runtime requires a novel action encoding (how to map any account to any region) that is scalable to arbitrary size of the action space (since the number of accounts keep growing). In Figure 4(l), we only test with a small switch with $3 \times 3$ ports, because standard A2C cannot encode or efficiently search the exponentially large action space when the number of ports grow beyond $10 \times 10$ (as described in §3.1). These tasks are examples were applying RL in realistic settings may require inventing new learning techniques (§3).

## 6. Conclusion

Park provides a common interface to a wide spectrum of real-world systems problems, and is designed to be easily-extensible to new systems. Through Park, we identify several unique challenges that require new algorithmic development in RL. The platform makes systems problems easily-accessible to researchers from the machine learning community so that they can focus on the algorithmic aspect of these challenges. We have open-sourced Park along with the benchmark RL agents and the existing baselines in https://github.com/park-project.

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

# Appendices

## A. Detailed descriptions of Park environments

We describe the details of each system environment in Park. Each description is structured to follow the problem background, MDP abstraction of the system interaction, the existing system-specific baseline heuristic approach, and how RL is suitable for the system problem.

**Adaptive video streaming.** The volume of video streaming has reached almost $60\%$ of all the Internet traffic (Sandvine, 2018). Streaming video over variable-bandwidth networks (e.g., cellular network) requires the client to adapt the video bitrate to optimize the user experience. In industrial DASH standard (Akamai, 2016), videos are divided into multiple chunks, each of which represents a few seconds of the overall video playback. Each chunk is encoded at several discrete bitrates, where a higher bitrate implies a higher resolution and thus a larger chunk size. For this problem, each MDP episode is a video playback with a particular network trace (i.e., a time series of network throughput). At each step, the agent observes the past network throughput measurement, the current video buffer size, and the remaining portion of the video. The action is the bitrate for the next video chunk. The objective is to maximize the video resolution and minimize the stall (which occurs when download time of a chunk is larger than the current buffer size) and the reward is structured to be a linear combination of selected bitrate and the stall when downloading the corresponding chunk. Prior adaptive bitrate approaches construct heuristic based on the buffer and network observations. For example, a control theoretic based approach (Yin et al., 2015) conservatively estimates the network bandwidth and use model predictive control to choose the optimal bitrate over the near-term horizon. In practice, the network condition is hard to model and estimate, making a fixed, hard-coded model-based approach insufficient to adapt to changing network conditions (Mao et al., 2017; Akhtar et al., 2018).

**Spark cluster job scheduling.** Efficient utilization of expensive compute clusters matters for enterprises: even small improvements in utilization can save millions of dollars at scale (Barroso et al., 2013). Cluster schedulers are key to realizing these savings. A good scheduling policy packs work tightly to reduce fragmentation (Verma et al., 2014), prioritizes jobs according to high-level metrics such as user-perceived latency (Verma et al., 2015), and avoids inefficient configurations (Ferguson et al., 2012). Since hand-tuning scheduling policies is uneconomic for many organizations, there has been a surge of interest in using RL to generate highly-efficient scheduling policies automatically (Mao et al., 2016; Chen et al., 2018; Mao et al., 2018).

We build our scheduling system on top of the Spark cluster manager (Zaharia et al., 2012). Each Spark job is represented as a DAG of computation stages, which contains identical tasks that can run in parallel. The scheduler maps executors (atomic computation units) to the stages of each job. We modify Spark's scheduler to consult an external agent at each scheduling event (i.e., each MDP step). A scheduling event occurs when (1) a stage runs out of tasks (i.e., needs no more executors), (2) a stage completes, unlocking the tasks of one or more of its children, or (3) a new job arrives in the system. At each step, the cluster has some available executors and some runnable stages from pending jobs. Thus, the scheduling agent observes (1) the number of tasks remaining in the stage, (2) the average task duration, (3) the number of executors currently working on the stage, (4) the number of available executors, and (5) whether available executors are local to the job. This set of information is embedded as features on each node of the job DAGs. The scheduling action is two-dimensional—(1) which node to work on next and (2) how many executors to assign to the node. We structure the reward at step $k$ as $r_k = -(t_k - t_{k-1})J_k$, where $J_k$ is the number of jobs in the system during the physical time interval $[t_{k-1}, t_k)$. Sum of such rewards penalize the agent in order to minimize the average job completion time. Park platform supports replaying an one-month industrial workload trace from Alibaba.

**SQL Database query optimization.** Queries in relational databases often involve retrieving data from multiple tables. The standard abstraction for combining data is through a sequential process that joins entries from two tables based on the provided filters (e.g., actor `JOIN` country `ON` actor.country_id = country.id) at each step. The most important factor that affects the query execution time is the order of joining the tables (Krishnan et al., 2018). While any ordering leads to the same final result, an efficient ordering keeps the intermediate results small, which minimizes the number of entries to read and process. Finding the optimal ordering remains an active research area, because (1) the total number of orderings is exponential in the number of filters and (2) the size of intermediate results depends on hard-to-model relationship among the filters. There have been a few attempts to learn a query optimizer using RL (Krishnan et al., 2018; Ortiz et al., 2018; Marcus et al., 2019).

Building the sequence of joins naturally fits in the MDP formulation. At each step, the agent observes the remaining tables to join as a query graph, where each node represents a table and the edges represent the join filters. The agent then decides which edge to pick (corresponds to a particular join) as an action. Park supports rewards from a cost model (a join cost estimate provided by commercial engines) and the final physical duration. In our implementation, we use Calcite (Begoli et al., 2018) as the query optimization framework, which can serve as a connector to any database management system (e.g., Postgres (PostgreSQL, 2019)).

**Network congestion control.** Congestion control has been a perennial problem in networking for three decades (Jacobson, 1988), and governs when hosts should transmit packets. Transmitting packets too frequently leads to congestion collapse (affecting all users) (Nagle, 1984) while over-conservative transmission schemes under-utilize the available network bandwidth. Good congestion control algorithms achieve high throughput and low delay while competing fairly for network bandwidth with other flows in the network. Various congestion control algorithms, including learning-based approaches (Dong et al., 2018; Yan et al., 2018; Jay et al., 2018), optimize for different objectives in this design space. It remains an open research question to design an end-to-end congestion control scheme that can automatically adapt to high-level objectives under different network condition (Shapira & Winstein, 2017).

We implement this enviroment using CCP (Narayan et al., 2018), a platform for expressing congestion control algorithms in user-space. At each step, the agent observes the network state, including the throughput and delay.[1] The action is a tuple of pacing rate and congestion window. The pacing rate controls the inter-packet send time, while the congestion window limits the total number of packets in-flight (sent but not acknowledged). We set our (configurable) action interval at 10 ms (suitable for typical Internet delays). Our reward function is adopted from the Copa (Arun & Balakrishnan, 2018) algorithm: `log(throughput) – log(delay)/2 – log(lost packets)`. This environment supports different network traces, from cellular networks to fixed-bandwidth links (emulated by Mahimahi (Netravali et al., 2015)).

---

[1]See Table 2 of Narayan et al. (2018) for full list.

**Network active queue management.** In network routers and switches, active queue management (AQM) is a fundamental component that controls the queue size (Athuraliya et al., 2001). It monitors the queuing dynamics and decides to drop packets when the queue gets close to full (Floyd & Jacobson, 1993). The goal for AQM is to achieve high throughput and low delay for the packets passing through the queue. Designing a strong AQM policy that achieves this high-level objective for a wide range of network condition can be complex. Standard methods — such as PIE (Hollot et al., 2001), based on PID control (Astrom et al., 2006) — construct a policy for a low-level goal that maintains the queue size at a certain level. In our setting, the agent observes the queue size and network throughput measurement; it then sets the packet drop probability. The action interval is configurable (default interval 10 ms; can also go down to per packet level control). The reward can be configured as a penalty for the difference between observed and target queue size, or a weighted combination of network throughput and delay. Similar to the congestion control environment, we emulate the network dynamics using Mahimahi with a wide range of real-world network traces.

**Tensorflow device placement.** Large scale machine learning applications use distributed training environments, where neural networks are split across multiple GPUs and CPUs (Mirhoseini et al., 2017). A key challenge for distributed training is how to split a large model across heterogeneous devices to speed up training. Determining an optimal device placement is challenging and involves intricate planning, particularly as neural networks grow in complexity and approach device memory limits (Mirhoseini et al., 2018). Motivated by these challenges, several learning based approaches have been proposed (Mirhoseini et al., 2017; 2018; Gao et al., 2018; Addanki et al., 2018).

We build our placement system on top of Tensorflow (Abadi et al., 2016). Each model is represented as a computational graph of neural network operations. A placement scheme maps nodes to the available devices. We formulate the MDP as an iterative process of placement improvement steps (Addanki et al., 2018). At each step, the agent observes an existing placement graph and tries to improve its runtime by updating the placement at a particular node. The state observation is the computation graph of a Tensorflow model, with features attached to each node which include (1) estimated node run time (2) output tensor size (3) current device placement (4) flag of the "current" node (5) flag if previously placed. The action places the current node on a device. Since the goal is to learn a policy that can iteratively improve placements, the reward $r_i = -(t_i - t_{i-1})$, where $t_i$ is the runtime of the placement at step $i$. Park supports optimizing placements for graphs with hundreds of nodes across a configurable number of devices. To speedup training, Park also provides a simulator for the runtime of a device placement (based on measurements from prior executions, see Appendix A4 in Addanki et al. (2018) for details).

**Circuits Design.** Analog integrated circuits often involve complex non-linear models relating the transistor sizes and the performance metrics. Common practice for optimizing analog circuits relies on expensive simulations and tedious manual tuning from human experts (Razavi, 2002). Prior work has applied Bayesian optimization (Lyu et al., 2018a) and evolution strategy (Liu et al., 2010) as general black-box parameter tuning tools to optimize the analog circuit design pipeline. Wang et al. (2019a;b) recently proposed to use RL to end-to-end optimize the circuit performance.

Park supports transistor-level analog circuit design (Razavi, 2002), where the circuit schematic is fixed and the agent decides the component parameters. For each schematic, the agent observes a circuit graph where each node contains the component ID, type (e.g., NMOS or PMOS) and static parameters (e.g., $Vth_0$). The corresponding action is also a graph in which each node must specify the transistor size, capacitance and resistance. Then, the underlying HSPICE circuit simulator (Synopsys, 2019) returns a configurable combination of bandwidth, power and gain as a reward. We refer the readers to Wang et al. (2019a) for more details.

**CDN memory caching.** In today's Internet, the majority of content is served by Content Delivery Networks (CDNs) (Nygren et al., 2010). CDNs enable fast content delivery by caching content in servers near the users. To reduce the content retrieval cost from a data center, CDNs aim to maximize the fraction of bytes served locally from the cache, known as the byte hit ratio (BHR) (Hasan et al., 2014). The admission control problem of CDN caching fits naturally to the MDP setting. At each step when an uncached object arrives in the CDN, the agent observes the object size, the time since the previous visit (if available) and the remaining CDN cache size. The agent then takes an action to admit or drop the uncached object. To maximize BHR, the reward at each step is the total byte hits since the last action (i.e., counting the size of cached objects served). Coupled with the admission policy is an eviction policy that decides which cached object to remove in order to make room for a newly admitted object. By default, our environment uses a fixed least-recently-used policy for object eviction. The environment also supports training an eviction agent together with the admission agent (e.g., via multi-agent RL). Our setup includes a real world trace with 500 million requests collected from a public CDN serving top-ten US websites (Berger, 2018).

**Multi-dim database indexing.** Many analytic queries to a database involve filter predicates (e.g., for query "SELECT COUNT(*) FROM TransactionTable WHERE state = CA AND day1 $\leq$ time $\leq$ day2", the filters are over state and time). Key to efficiently answering such range queries is the database index — the layout in which the underlying data is organized (e.g., sorted by a particular dimension). Many databases choose to index over multiple dimensions because analytics queries typically involve filters over multiple attributes (IBM). A good index is able to quickly return the query result by minimizing the number of points it scans. We found empirically that a well-chosen index can achieve query performance three orders of magnitude faster than one that is randomly selected. In practice, choosing a good index depends on the underlying data distribution and query workload at runtime; therefore, many current approaches rely on routine manual tuning by database administrators.

We consider the problem of selecting a multi-dimensional index from an RL perspective. We target grid-based indexes, where the agent is responsible for determining the size of the cells in the grid. We found that this type of index is competitive with traditional data structures, while offering more learnable parameters. At each step of our MDP formulation, the database receives a new set queries to run, and the agent has the opportunity to modify the grid layout. The observation consists of both the dataset (i.e., list of records in the database) and queries (i.e., a list of range boundaries for each attribute) that have arrived since the previous action. The environment then (1) samples a workload from a distribution that changes (slowly) over time, (2) uses it to evaluate the agent-generated index on a real column-oriented datastore, and (3) reports the query throughput (i.e., queries per second) as the agents reward. Our environment uses a real dataset collected from Open Street Maps (OpenStreetMap contributors, 2019) with 105 million records, along with queries sampled from a set of relevant business analytic questions. In this setup, there are more than 7 trillion possible grid layouts that the agent must encode in its action space.

**Account region assignment.** Social network websites reduce access latency by storing data on servers near their users. For each user-uploaded piece of content, the service providers must decide which region to serve the content from. These decisions have a multitude of tradeoffs: storing a piece of content in many regions incurs increased storage cost (e.g., from a cloud service provider), and storing a piece of content in the "wrong" region can substantially increase access latency, diminishing the end user's experience (Alicherry & Lakshman, 2012).

To faithfully simulate this effect, our environment includes a real trace of one million posts created on a medium-sized social network over eight months from eight globally distributed regions. Park supports two variants of the assignment task. First, the agent chooses a region assignment when a new piece of content is initially created. At each content creation step, the observation includes the language, outgoing links, and posting user (anonymized) ID. The action is one of the eight regions to store the content. The reward is based on the fraction of accesses from within the assigned region. This variant can be viewed as a contextual multi-armed bandit problem (Lu et al., 2010). The second variant is similar to the first one, except that the agent has the opportunity to migrate any content to any region at the end of each 24 hour time period. The action space spans all possible mappings between the users and the regions. In this case, the agent must balance the cost of a migration against the potential decrease in access latency.

**Server load balancing.** In this simulated environment, an RL agent balances jobs over multiple heterogeneous servers to minimize the average job completion time. Jobs have a varying size that we pick from a Pareto distribution (Grandl et al., 2016) with shape 1.5 and scale 100. The job arrival process is Poisson with an inter-arrival rate of 55. The number of servers and their service rates are configurable, resulting in different amounts of system load. For example, the default setting has 10 servers with processing rates ranging linearly from 0.15 to 1.05. In this setting, the load is 90%. The problem of minimizing average job completion time on servers with heterogeneous processing rates does not have a closed-form solution (Harchol-Balter & Vesilo, 2010); a widely-used heuristic is to join the shortest queue (Daley, 1987). However, understanding the workload pattern can give a better policy; for example, one strategy is to dedicate some servers for small jobs to allow them finish quickly even if many large jobs arrive (Feng et al., 2005). In this environment, upon each job arrival, the observed state is a vector $(j, s_1, s_2, ..., s_k)$, where $j$ is the incoming job size and $s_k$ is the size of queue $k$. The action $a \in \{1, 2, ..., k\}$ schedules the incoming job to a specific queue. The reward $r_i = \sum_n [\min(t_i, c_n) - t_{i-1}]$, where $t_i$ is the time at step $i$ and $c_n$ is the completion time of active job $n$.

**Switch scheduling.** Switch scheduling poses a matching problem that transfers packets from the incoming ports to the outgoing ports (McKeown, 1999; Shah & Wischik, 2006; Maguluri & Srikant, 2016). This abstracted model is ubiquitous in many real world systems, such as datacenter routers (Giaccone et al., 2002) and traffic junctions (Hunter et al., 1997). At each step, the scheduling agent observes a matrix of queue lengths, with element $(i, j)$ indicating the packet queue from input port $i$ to output port $j$. The matching action is bijective — no two incoming packets shall pass through the same output ports. Notice that in a switch with $n$ input/output ports, the action space is the $n!$ possible bijection matchings.[2] After each scheduling round, one packet is transferred per each input/output port pair. The goal is to maximize switch throughput while minimizing packet delay. The optimal scheduling policy for this problem is unknown and is conjectured to depend on the underlying traffic pattern (Shah & Wischik, 2006). For example, the max weight matching policy empirically performs well only under high load (Maguluri & Srikant, 2016). Adapting the scheduling policy under dynamics load to optimize an arbitrary combination of throughput and delay is challenging.

## B. Experiment setup

This section details the experiment setup for benchmarking existing RL algorithms in Park. We show the result of the benchmarks in Figure 4.

---

[2]Typical routers can have 144 ports (Greenberg et al., 2009).

## B.1. RL algorithms

We follow the standard implementations of existing RL algorithms in OpenAI baselines (Dhariwal et al., 2017). Specifically, A2C (Mnih et al., 2016) uses separated policy and value network and it has training batch of size 64. For discrete-action environments, A2C explores using an entropy term in policy loss (Mnih et al., 2016; Wu & Tian, 2017), with the entropy factor linearly decay from 1 to 0.01 in 10,000 iterations. For continuous-action environments, the policy network outputs the mean of a Gaussian distribution. The variance is controlled by an external factor that decays according to the same schedule as the discrete case. In Policy Gradient (PG) (Sutton et al., 1999), we rollout 16 parallel trajectory and we use a simple time-based baseline averaging the return across the trajectories. DQN (Mnih et al., 2015) employs a replay memory with size 50,000 and updates the target Q network every 100 steps. DDPG (Lillicrap et al., 2015) uses a small replay memory with 2048 objects and updates the target networks every 1000 steps.

For feed forward networks, we use simple fully connected architecture with two hidden layers of 16 and 32 neurons. For recurrent neural networks, we use LSTM with 4 hidden layers. We use graph convolution neural networks (GCNs) (Kipf & Welling, 2016) to encode the states that involve a graph structure. In particular, we modify the message passing kernel in Spark scheduling and Tensorflow device placement problems. The kernel is $\mathbf{e}_v \leftarrow g\left[\sum_{u \in \xi(v)} f(\mathbf{e}_u)\right] + \mathbf{e}_v$, where $\mathbf{e}$ is the feature vector on each node, $f$ and $g$ are non-linear transformatio implemented by feed forward networks, $\xi(\cdot)$ denotes the child nodes. When updating the neural network parameters, we use Adam (Chilimbi et al., 2014) as the optimizer. The non-linear activation function is Leaky-ReLU (Nair & Hinton, 2010). We do not observe significant performance change when changing the hyperparameter settings.

## B.2. Environment
### configuration and comparing baselines

**Adaptive video streaming.** We train and test the A2C agent on the simulated version of the video streaming environment since the interaction with real environment is slow. However, the learned policy can generalize to a real video environment if the underlying network conditions are similar (Mao et al., 2017). We compare the learned A2C policy against two standard schemes. The "buffer-based" heuristic switches the bitrates purely based on the current playback buffer size (Huang et al., 2014). "robustMPC" uses a model predictive control framework to decide the bitrate based on a combination of the current buffer size and a conservative estimate of the future network throughput (Yin et al., 2015). We use the default parameters in the baseline algorithm from their original paper (Yin et al., 2015).

**Spark cluster job scheduling.** The benchmark experiment is on a cluster of 50 executors with a batch of 20 Spark jobs from the TPC-H dataset (tpch). During training in simulation, we sample 20 jobs uniformly at random from all available jobs. We test on a real cluster with the same setup and unseen job combinations. The "fair" scheduler gives each job an equal fair share of the executors and round-robins over tasks from runnable stages to drain all branches concurrently. The "optimal weighted fair" scheduler is carefully-tuned to give each job $T_i^\alpha / \sum_i T_i^\alpha$ of the total executors, where $T_i$ is the total work of each job $i$ and $\alpha$ is a tuning factor. Notice that $\alpha = 0$ reduces to a simple fair scheme and $\alpha = 1$ reduces to a weighted fair scheme based on job size. We sweep through $\alpha \in \{-2, -1.9, ..., 2\}$ for the optimal factor.

**SQL Database query optimization.** We train and test a DQN agent on a cost model implemented in the open source query optimization framework, Calcite. This provides an estimate of the number of records that would have to be processed when we choose an edge in the query graph (apply a Join), and how long it would take to process them based on the hardware characteristics of the system. The cost model is based on the non-linear cost model ('CM2') described by (Krishnan et al., 2018), where the non-linearity models the random access memory constraints of a physical system. The training set, and test set, are generated from 113 queries in the Join Order Benchmark (Leis et al., 2015), with a $50\%$ train-test split. We use the following baselines from traditional database research to compare against the RL approach. *(1) Exhaustive Search:* For a given cost model, we can find the optimal policy using a dynamic programming algorithm (Exhaustive Search) and all our results are presented relative to this ($-1.00$ means the plan was as good as Exhaustive Search plan). *(2) Left Deep Search:* Is a popular baseline in practice since it finds the the optimal plan in a smaller search space (only considering join plans that form a left deep tree (Krishnan et al., 2018)) making it computationally much faster than Exhaustive Search.

**Network congestion control.** We train and test the A2C agent in the centralized control setting (a single TCP connection) on a simple single-hop topology. We used a 48Mbps fixed-bandwidth bottleneck link with 50ms round-trip latency and a drop-tail buffer of 400 packets (2 bandwidth-delay products of maximum size packets) in each direction. For comparison, we run TCP Vegas (Brakmo & Peterson, 1995). Vegas attempts to maintain a small number of packets (by default, around 3) in the bottleneck queue, which results in an optimal outcome (minimal delay and packet loss, maximal throughput) for a single-hop topology without anycompeting traffic. "Confined search space" means we confine the action space of A2C agent to be only within $0.2$ and $2\times$ of the average action output from Vegas.

**Network active queue management.** We train and test the agent on a 10Mbps fixed-bandwidth bottleneck link with 100ms round-trip latency where there are 5 competing TCP flows. The agent examines the state and takes an action every 50ms. We configure the reward to be the current distance from the target queuing delay (20ms). As a comparison, we run "PIE" (Hollot et al., 2001), a classic PID control scheme, with the same target queuing delay.

**Tensorflow device placement.** We consider device placement optimization for a neural machine translation (NMT) model (Bahdanau et al., 2014) over two devices (GPUs). This is a popular language translation model that has an LSTM-based encoder-decoder and attention architecture to translate a source sequence to a target sequence. The training is done over a reliable simulator (Addanki et al., 2018) to quickly obtain run-time estimates given a placement configuration. In the "Single GPU" heuristic, all ops are co-located on the same device, which is optimal for models that can fit in a single device and which do not have significant parallelism in their structure. Scotch (Pellegrini, 2007) is a graph partitioning based heuristic that takes as input both the computational cost of each node and the communication cost along each edge. It then outputs a placement that minimizes total communication cost, while load balancing computation across the devices to within a specified tolerance. The human expert places each LSTM layer on a different device as recommended by Wu et al. (Bahdanau et al., 2014). PG-LSTM (Mirhoseini et al., 2017) embeds the graph model as a sequence of node features, and uses an LSTM to output the corresponding placement for each node in the sequence. The PG-GCN (Addanki et al., 2018) on the other hand, uses a graph neural network (Bronstein et al., 2017; Hamilton et al., 2017) for embedding the model, and represents the policy as performing iterative placement improvements rather than outputting a placement for all the nodes in one shot.

**Circuits Design.** The benchmark trains and tests on a fixed three-stage transimpedance amplifier analog circuit. "BO" is a simple Bayesian optimization approach to tune the model parameter. "MACE" is a prior work based on acquisition function ensemble (Lyu et al., 2018b). "ES" stands for evolutional strategy approach (Salimans et al., 2017). "NG-RL" is the short of non-grach Reinforcement Learning in which we do not involve graph informantion in the optimzation loop. "GCN-RL" is the Reinforcement Learning with graph convolutional neural networks. From the results, we can observe that "GCN-RL" could consistently achieve higher Figure of Merits (FoM) value than other methods. Comparing to "NG-RL", "GCN-RL" has higher FoM value and also faster convergence speed, which indicates the critical role of the graph information.

**CDN memory caching.** We train and test A2C on several synthetic traces (10000 requests long) produced by an open-source trace generator (Berger et al., 2017). We consider a small cache size of 1024KB for the experiment. The LRU heuristic always admits requests, with stale objects evicted based on the last recently used (LRU) policy. Offline optimal uses dynamic programming to compute the best sequence of actions, with the knowledge of future object arrivals.

**Multi-dim database indexing.** We train and test on a real in-memory column-store, using a dataset from Open Street Maps (OpenStreetMap contributors, 2019), comprised of 105 million points, each with 6 attributes. The dataset is unchanged across all steps. The query workload shifts continuously between different query distributions, completing a full shift to a new distribution every 20 steps. At each step, the agent observes the previous workload and produces a parametrization of the grid index that is tested on the next workload. We use a batch size of 1, and the environment is terminal at every state (i.e., the discount factor $\gamma$ is 0).

We heavily restrict the state and action space to make this environment tractable. The agent does *not* observe the underlying data, since the dataset does not change; it observes only the query workload. Each workload consists of 10 queries, each with two 6-dimensional points to specify the query rectangle, producing a 120-dimensional observation space. Each query coordinate is scaled to $[0,1]$, relative to the range of the corresponding attribute in the OSM dataset. If an attribute is not present in the range filter, the query coordinates for that dimension are 0 and 1. For the agent's action, we fix an ordering of dimensions that we have found to work well empirically; the agent is responsible solely for determining the number of columns along each dimension in the grid, which is a 4-dimensional action space. The baseline is a fixed layout that is run on the same workloads as the agent, tuned roughly by hand to produce low running times on the *entire* sequence of workloads. The baseline layout uses the same dimension ordering that was fixed for the agent and is not re-optimized for each new workload.

**Account region assignment.** The setup for this experiment follows the first variant of the assignment task outlined in Appendix A, in which the agent has to assign newly created accounts to one of eight regions. Local heuristic is a simple baseline that assigns an account directly to the region it was created in. The Thompson sampling (Chapelle & Li, 2011) approach uses a random forest model comprising of 100 trees. We train and test DQN over the real trace of one million posts included with Park.

**Server load balancing.** In this experiment we consider the setup as described in Appendix A, with 10 heterogenous servers. The A2C (Mnih et al., 2016) learning approach is elaborated in Appendix B.1; 'grad clip' refers to gradient clipping, in which we normalize the policy gradient by its l2 norm when the l2 norm is over 10. The greedy heuristic assigns each incoming job to that queue having the lowest queue size to processing rate ratio.

**Switch scheduling.** We consider scheduling in a crossbar switch (Appendix A) with 3 input ports and 3 output ports. Time is discretized for simplicity. Traffic between each port pair $(i,j)$ is generated according to a Bernoulli process, with rate given by the $(i,j)$-th entry of a random bistochastic traffic matrix. The load of the system (i.e., the row and column sums of the traffic matrix) is set to 90%. MWM, or Max-Weight-Matching (Shah & Wischik, 2006), is a well-known scheduling policy that forwards packets at each time-step according to the maximum weighted matching on the bipartite graph between the set of input and output ports. The weight of each edge $(i,j)$ on the bipartite graph is set equal to the size of the virtual-output queue (VOQ) $j$ at input port $i$ (Shah & Wischik, 2006). For a parameter $\alpha > 0$, MWM-$\alpha$ refers to an analogous policy where the weight of edge $(i,j)$ on the bipartite graph is set equal to the size of VOQ $j$ at input port $i$ raised to the power $\alpha$.