# OpenReview forum: "Park: An Open Platform for Learning Augmented Computer Systems"
_ICML.cc/2019/Workshop/RL4RealLife — RL4RealLife 2019_

### Official Review · AnonReviewer1 · 2019-05-24
**A useful and timely software platform for RL experiments with real systems**

**Rating:** 5
**Confidence:** 4

**Review:**

==============
== Summary ==
==============
This paper develops Park, an open source software platform that can facilitate easy application and testing of RL approaches to a variety of systems problems. Park standardizes the interface between RL algorithm and a real-world system with the goal of obscuring the low level complexity of the system itself. The supported environment types in Park are quite general, capturing systems problems like scheduling in a datacenter, network congestion control, and database query optimization. Critical to this workshop, Park exposes several challenges of using RL on real systems that are not well accounted for in existing models. For instance, the state-action space size can change as a given system changes over time. The paper overviews the features of Park and conducts experiments benchmarking typical RL algorithms on several of the tasks provided in Park.

==============
== Comments ==
==============
The paper is well written and presents an important contribution: Park offers a new tool for RL practictioners to scale methods beyond typical simulated environments and onto tasks that necessarily contain the complexity of the real world. The release of such a system will be of interest to many in the RL community, as it will both raise new research challenges regarding appropriate scale and robustness to real world tasks, and hopefully encourage new applications and algorithmic insights to meet the needs of these systems problems. Section 4 of the paper highlights challenges of running RL algorithms in these realistic systems, which is particularly appropriate for the topic of the meeting. Overall, this work is highly relevant for the workshop.

The code is well documented and easy to follow, and is available through github, but I ran into a few errors when running the unit tests.

==============
== References ==
==============
- For the high sample complexity of model-free approaches, I would suggest citing something with more supporting evidence, such as the recent work by Sun et al. (2019).


==============
== References ==
==============
[Background]
- "making an action for early part of an query requires" --> "choosing an action for an early part of a query requires"
- "have fast control loop with a clear" --> "have a fast control loop with a clear"
- "it is hard to directly to make one-shot decisions for partial" --> "it is hard to directly make one-shot decisions for a partial"
- "we provide simulated environment" --> "we provide a simulated environment"
- "For example, in memory caching environment" --> "For example, in a memory caching environment"

[Environments]
- "For state space involving graph structure" --> "For state spaces involving graph structure"
- Having trouble parsing: "As a sanity check, the performance of learned policy in these environments all improve over time"
- Also having trouble parsing: "However, in many scenarios, the off-the-shelf RL algorithms only perform well when we largely simply the settings or they struggle to outperform the existing deployed heuristics."

==============
== References ==
==============
Sun, Wen, et al. "Model-based reinforcement learning in contextual decision processes: PAC bounds and exponential improvements over model-free approaches." COLT 2019.

---

### Official Review · AnonReviewer2 · 2019-05-25
**Park platform for real and simulated RL**

**Rating:** 5
**Confidence:** 4

**Review:**

The paper presents Park an open source platform for benchmarking RL algorithms on real tasks from computer systems.
The paper describes the tasks (also nicely summarized in a Table) and the workings of the system (such as the fact
that unlike OpenAI gym, it makes more sense for the simulator to step the agent), as well as several challenges
they have found while benchmarking off-the-shelf RL algorithms on the Park workloads. It should not be surprising
that domain specific heuristics have been found hard to beat. These have been developed with safety of the system
in mind and when an RL agent takes an unsafe action it can quickly end up in a state it cannot recover from.

I think this paper and the platform will play a significant role in advancing RL research on real problems.

One thing I did not understand from the paper is what happens when real data is being used and the RL agent wants
to take an action that is different from the action in the log. How does Park decide what's the next state?

Pros:
- Open Platform
- Interesting set of problems for an RL environment (real + simulated with realistic simulator)
- Clear explanation of the challenges that RL algorithms run into.

Cons:
- The fixed featurization can be problematic as it couple the task with a specific definition of state which might turn out to be inadequate.
It should at least be possible to have different plugins for getting the state of the system.
- It is possible that extensive use of Park benchmarks can lead to overfitting. It would be nice to
to have a leaderboard where you can upload your RL agent e and it gets run on held out data from the same task.

---

### Decision · Program_Chairs · 2019-05-28

Accept